# PASS: An ImageNet replacement for self-supervised pretraining without humans

**Yuki M. Asano**     **Christian Rupprecht**     **Andrew Zisserman**     **Andrea Vedaldi**

Visual Geometry Group, University of Oxford
{yuki,chrisr,az,vedaldi}@robots.ox.ac.uk

## Abstract

Computer vision has long relied on ImageNet and other large datasets of images sampled from the Internet for pretraining models. However, these datasets have ethical and technical shortcomings, such as containing personal information taken without consent, unclear license usage, biases, and, in some cases, even problematic image content. On the other hand, state-of-the-art pretraining is nowadays obtained with unsupervised methods, meaning that labelled datasets such as ImageNet may not be necessary, or perhaps not even optimal, for model pretraining. We thus propose an unlabelled dataset **PASS: Pictures without humAns for Self-Supervision**. PASS only contains images with CC-BY license and complete attribution metadata, addressing the copyright issue. Most importantly, it contains *no images of people at all*, and also avoids other types of images that are problematic for data protection or ethics. We show that PASS can be used for pretraining with methods such as MoCo-v2, SwAV and DINO. In the transfer learning setting, it yields similar downstream performances to ImageNet pretraining even on tasks that involve humans, such as human pose estimation. PASS does not make existing datasets obsolete, as for instance it is insufficient for benchmarking. However, it shows that model pretraining is often possible while using safer data, and it also provides the basis for a more robust evaluation of pretraining methods. Dataset and pretrained models are available[1].

## 1  Introduction

The development of modern machine learning could not have happened without the availability of increasingly large and diverse research datasets. Consider computer vision for example: algorithms were initially developed using small datasets collected in laboratory conditions and, as a consequence, almost no method worked in the real world. This cycle was broken only once researchers adopted datasets such as PASCAL VOC, MS COCO and ImageNet, which are collections of images sampled from the Internet. These collections are not only much larger than prior datasets, but they also provide a much better representation of the statistics of natural images because they arise from the real world. Because of this, algorithms became vastly more robust and were able to better generalize to the real world. Furthermore, these datasets have acquired fundamental scientific functions as well: they allow reproducible and quantitative comparison of algorithms and they enable researchers to efficiently build on each other's work.

However, for all their benefits, these datasets have technical, ethical and legal shortcomings.

One issue is *copyright*. While some datasets contain images explicitly licensed for reuse (e.g. Creative Commons), some do not. Some national laws contain copyright exceptions or otherwise allow the

---

[1] https://www.robots.ox.ac.uk/ vgg/research/pass/

35th Conference on Neural Information Processing Systems (NeurIPS 2021) Track on Datasets and Benchmarks.

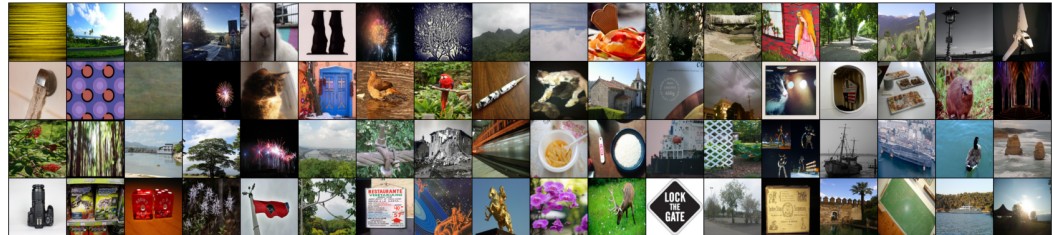

Figure 1: **PASS : Pictures without humAns for Self-Supervision.** We propose a new dataset of 1.28M Internet images with CC-BY license that do not contain humans or body parts at all. The dataset is collected randomly without using search engines and has no labels, which makes it particularly suitable for self-supervised learning. We show that this dataset can largely replace ImageNet for the purpose of model pretraining. While this does *not* make ImageNet obsolete, it does remove the need for using ImageNet for one of its most common applications. Individual image attributions for this figure are given in the Appendix.

usage of copyrighted works for research and/or for training machine learning models, but this is not a universal, nor always firmly established, fact.

Another issue is *data protection*. The vast majority of images are collected by humans for human consumption in human-populated areas. As a natural consequence, a large fraction of Internet images contain people. Because it is nearly impossible to obtain consent for all the people in these images, this data is collected without consent [8, 90]. This is an ethical issue as well as a legal one, as personal data is protected by legislation such as the EU and UK General Data Protection Regulation (GDPR). Data protection legislation may still allow the usage of such data for research purposes, but this generally requires the data to be minimal, i.e. required for the specific research one is conducting.

While the copyright issue can sometimes be addressed by choosing liberally-licensed images, the data protection issue is much more challenging. The key question here is whether computer vision can simply avoid having any images containing people. When the goal is to extract information about people (e.g., pose recognition), the answer is obviously no. However, datasets such as ImageNet are often used for model pretraining even if the final application is not about people at all. In this case, it is legitimate to ask if pretraining could be done on data that does not contain people.

Note that blurring people as recently done in [91], while helpful, is not enough to remove privacy concerns; for instance, based on GDPR such images are still personal data insofar as the blurred individual can be recognized due to e.g. contextual cues, which can also be picked up by reverse image search engines [70]. We look instead at whether people can be excluded *completely*.

Rather than just filtering ImageNet, however, we take a different route. Many datasets such as ImageNet, which were designed for supervised object classification, contain undesirable *biases*. For example, [8] highlights harmful depictions in the popular ImageNet 2012 split, and its collection and annotation processes have been criticized for leading to stereotyped and problematic depictions of categories [8, 75, 78, 79]. For ImageNet, we therefore identify as main sources of bias its collection by scraping with search engines, and its selection of 1000 labels. On the other hand, we note that the current state-of-the-art model pretraining uses self-supervised learning (SSL) and thus does *not* require labels at all. Motivated by this, we thus consider forming a dataset *without using labels*, significantly increasing diversity and removing the search engine selection bias. Because we *remove images with humans*, we further significantly reduce the risk of including contextual biases linked to the appearance of people. Furthermore, due to its more random and unsupervised nature, this dataset also serves as a better benchmark for SSL to study scaling to natural images that are not curated to a pre-defined set of class labels, addressing a technical shortcoming of current evaluations.

Concretely, our first contribution is **PASS**, a large collection of images (1.28M) excluding humans (along with other identifiable information such as license plates, signatures, or handwriting and NSFW images). We do so by starting from a large-scale (100 million random flickr images) dataset—YFCC100M [81]—meaning that the data is better randomized[2] and identify a 'safer' subset within it.

---

[2]Of course, this does not remove all biases, as we discuss in the limitations section.

We also focus on data made available under the most permissive Creative Common license (CC-BY) to address copyright concerns.

Given this data, we then conduct an extensive evaluation of SSL methods, discussing performance differences when these are trained using ImageNet and **PASS**. Compared to ImageNet, there are three essential differences with the **PASS** dataset: the lack of class-level curation and search; the lack of 'community optimization' on this dataset; and, of course, the lack of humans. We study via further ablation the contribution of these effects.

We find that: (i) self-supervised approaches such as MoCo, SwAV and DINO train well on our dataset, yielding strong image representations; (ii) excluding images with humans during pretraining has almost no effect on downstream task performances, even if this is done in ImageNet; (iii) in 8/13 frozen encoder evaluation benchmarks, performance of models trained on **PASS** yield better results than pretraining on ImageNet, ImageNet without humans, or Places205, when transferred to other datasets; and for finetuning evaluations, such as detection and segmentation, **PASS** pretraining yields results within $\pm 1\%$ mAP and AP50 on COCO. (iv) Even on tasks involving humans, such as dense pose prediction, pretraining on our dataset yields performance on par with ImageNet pretraining.

## 2  Related Work

**Image datasets for model pretraining.**  By far the most widely used pretraining dataset is ImageNet ILSVRC12 (IN-1k) [22], containing 1.28M images covering 1000 object categories. This is followed by MIT-Places 205 and 365 datasets [97, 99], containing 2M and upto 10M images with labels, respectively, Webvision [59] containing the same labels as IN-1k, as well as Taskonomy [92], containing 4M images with scene attributes.

YFCC-100M [81] is a much larger dataset with 99M images with licence information and other metadata. OpenImages [6, 54, 56] contains 6M images and 20K classes. These datasets have been further combined and relabeled to yield further datasets, including Tencent ML-images [86] and Conceptual Captions [76].

When measured on downstream task performance, supervised pretraining on ImageNet (IN-1k) has nowadays been surpassed by unsupervised pretraining, but often these methods use the same images for both pretraining and evaluation (with labels). We argue that SSL methods should instead be trained on more diverse and less-curated datasets, so as to disentangle the performance gains from the potential requirements on the underlying data. This is particularly true if ImageNet is *also* used for evaluation.

**Bias and privacy.**  Biases contained in datasets have been investigated in various works [8, 9, 20, 25, 42, 46, 68, 78, 90, 94] and in particular its annotation practices [1, 7, 34, 48, 63, 65, 71, 82, 83]. We focus specifically on ImageNet as this is the defacto standard dataset used for pretraining representations[3]. Notably, [20] shows that the larger version of ImageNet had stereotypes/slurs as class labels, and further was biased with regards to gender-biased depictions [25], which led to the removal of the person categories from the dataset [90].

More recently, Birhane and Prabhu [8] finds biased and NSFW images in ImageNet LSVRC-12 and criticises its labelling process. This paper and its subsequent blog-post [70] also outline problematic categories, such as the 'bikini' class. Another issue is that ImageNet, as well as other datasets, contain biases inherited from the search engines used to collect them, which sometimes results in racist, sexualized, skewed or stereotyped representations [42]. These biases are passed down to models trained on this data, e.g. as shown in the image completion work [78], or face upsampling [62]. Furthermore, the labels included in ImageNet are also imperfect: when [71] reproduced ImageNet's annotation process, they noted highly-variable results yielding up to 11%–14% drops in accuracy. For a more in-depth review of datasets, their collection practises and current issues we refer to Paullada et al. [68].

In response to some of these issues, a version of ImageNet with blurred faces has now been published [91], in which the authors show that this only marginally affects object classification perfor-

---

[3]Although these issues are likely to be present in most other web images datasets too: following the methodology of [8], we easily find several pornographic and troubling images even in Places205 [98]. Better filtering for such content is necessary.

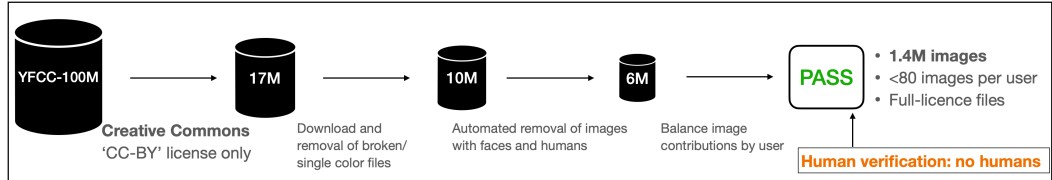

Figure 2: Dataset generation pipeline. Starting from 100M images in YFCC, we select 6M images that do not contain humans. This dataset is further reduced to the size of 1.4M by balancing user contributions and is finally manually checked.

mance. However, as [70] points out, this does not eliminate the problem, as even after blurring faces, reverse image search can be used to retrieve the original picture, thus leaving the privacy concerns unsolved.

In this work, we focus on alleviating some of these privacy and bias issues by constructing a dataset that is completely free of humans, does not rely on search engines or labels, and only contains images with a permissive license (CC-BY) and attribution information.

**Self-supervised learning.** Starting from early works such as [47, 67], self-supervised representation learning methods methods are nowadays mainly based on clustering [3, 11, 35, 36, 58] and/or noise-contrastive instance discrimination [14, 29, 38, 41, 44, 64, 88]. In particular, MoCo [44] has sparked several further implementations [4, 15, 51, 58, 77, 89, 100] and therefore we use this model for most of the experiments in this paper. Several papers [30, 53, 95] also compare SSL methods in detail and find among other results that, downstream task performance is dependent on the pretraining dataset, and that self-supervised methods struggle on fine-grained classification tasks compared to their supervised counterparts. We therefore evaluate various distributional splits of our dataset and evaluate also on finegrained datasets.

# 3 PASS: images minus people

The PASS dataset is obtained as a subset of the YFCC-100M dataset [81]. The latter is freely available on the AWS public datasets page [74], and has complete metadata that includes the individual licenses and creators, thus satisfying our requirement of CC-BY. In case this dataset ceases to exist on the public datasets of AWS, we have made necessary backups, for details see the 'Datasheets of Datasets' section in the Appendix.

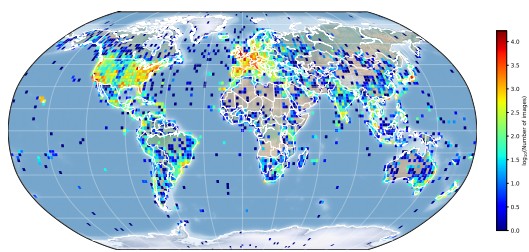

Figure 3: Geographic diversity: 552K of 1.4M images contain geo-locations.

**Collection modality.** PASS was obtained as follows (see also Fig. 2). First, the YFCC-100M metadata was used to select images with CC-BY licence, which left us with 17M images. Once, these images were downloaded, corrupted or single-color images[4] were removed (leaving 10M). Pretrained RetinaFace [23] and Cascade-RCNN (3x) [10] models were used to filter out images containing human faces and humans (6M). All details are provided in the Appendix.

In YFCC-100M, the distribution of images per photographer is highly skewed. Thus, to increase dataset diversity, we set a maximum threshold of 80 images per photographer and uniformly sampled 1.4M images. Finally, these images were submitted for human labelling[5].

**Manual Filtering.** The annotators were asked to identify images that contain people or body parts, as well as personal information such as IDs, names, license plates, signatures etc. Additionally, the annotators were asked to flag images with problematic content such as drugs, nudity (mostly included

---

[4]For a surprisingly large fraction of the images all pixels have the same color.

[5]Prior to human verification we further checked the most highly ranked 1K images according to a NSFW classifier [57] for possible harmful content and found none.

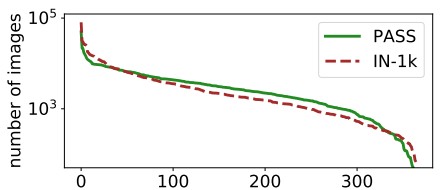

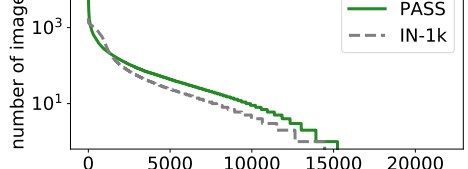

(a) **Distribution of places.** As determined by a Places-365 pretrained ResNet-50.

(b) **Distribution of objects.** As determined by a ImageNet-21k pretrained ResNet-50x3 classifier.

Figure 4: **Larger diversity than ImageNet-1k.** Compared to the curated ImageNet-1k dataset, our dataset has longer tails.

Table 1: Downstream evaluation tasks and datasets. (⚥) Indicates presence of 'person' annotations or human-oriented downstream tasks.

| Frozen | Clustering: IN-1k, ObjectNet, Places205, Flowers |
| | Linear probing: IN-1k, ObjectNet, Places205, Flowers, CIFAR-10 (⚥), Herbarium2019 |
| | Low-shot classification: Pascal VOC07 (⚥), Places205, Flowers |
| Finetune | Object detection: COCO17 (⚥), Pascal VOC07+12 (⚥) |
| | Object segmentation: COCO17 (⚥), LVIS (⚥) |
| | Dense pose detection (⚥): COCO17 |
| | Human keypoint detection (⚥): COCO17 |

in the "no people" rule), blood and other offensive content. The annotations were performed over the course of three weeks by an annotation company whose annotators were paid 150% of the minimum wage. During the manual annotation process, around 2% of the images were flagged and subsequently removed. From the remaining images (1.46M) we further removed duplicates (see Appendix C.3) and randomly selected a subset with approximately the same size as IN-1k (1.440.191 images).

**Statistics.** We first provide some descriptive results of our dataset. Using the meta-data provided, we plot the GPS location of the images in Fig. 3. We find that the subset of the dataset that contains GPS coordinates covers a wide area of the world, but is focused on western countries such as the US, Europe and Japan. This likely reflects the skew of the Flickr user database and is a form of bias that can limit generalisation capabilities [21], as for example images taken in less developed countries might be uploaded by tourists that take images of stereotypes [75, 84]. With this we wish to stress that the **PASS** dataset is by no means bias-free, and its misunderstanding as such might lead to spurious claims of fairness. While the dataset comes with some textual annotations such as tags and descriptions, these are discarded to avoid further biases [83] since they are not necessary for self-supervised learning.

In Fig. 4 we show the results of running pretrained classification models on our dataset and compare the output diversity against IN-1k. Note that in Fig. 4b, the distribution for IN-1k objects should technically be zero for classes after 1000, as IN-1k is a subset of IN-21k. We attribute this deviation to the quality of the model—classifiers with a very high number of classes (here: 21,000) are still very noisy. Nonetheless, for a relative comparison of two datasets, even a noisy classifier is sufficient. We find that, compared to IN-1k, our dataset contains more variety in terms of places and objects, as shown by the higher tail of the distributions.

## 4 Experimental setup

We evaluate **PASS** by pretraining models on the **PASS** data using self-supervision and then testing the quality of the resulting models on a range of downstream tasks outlined in Table 1.

**SSL pretraining methods.** We pretrain models using MoCo-v2 [16] as it has become a staple SSL method, can be extended in various ways [4, 15, 51, 58, 77, 89, 100] and, with updates to the learning schedule and most recently its implementation of the loss [17] remains state-of-the-art in feature learning. This makes it an ideal candidate to probe the pretraining performance of our dataset. We therefore conduct most of our experiments using the MoCo-v2 public implementation with a 200

epoch pretraining schedule. We further train and evaluate models trained with SwAV [12], another state-of-the-art representation learning method, but one which is based on unsupervised clustering. Finally, we also pretrain using DINO [13], a recent method that is tailored to vision transformer [24] SSL.

For downstream evaluation, we set all training parameters as proposed in existing literature (which, in practice, means they are tuned for IN-1k). Specific tuning for each pretraining dataset would probably improve performance. For pretraining MoCo-v2 and MoCo-v2 CLD, we follow the official implementation and adhere to the 200 epoch schedule. We use the 200 epoch configuration with multi-crop schedule for SwAV which takes around double the GPU hours as MoCo's 200 epochs. For DINO, we follow their 100 epoch training code, which also includes multi-crop. Further experimental details are given in the Appendix.

**Pretraining Data.** Using MoCo-v2, we compare **PASS** against other datasets for pretraining, such as: **(a)** IN-1k: ImageNet-2012 split [22], the current standard for SSL pretraining; **(b)** IN-1k$^*$: ImageNet-2012 with humans and human parts removed, as an ablation of IN-1k; **(c)** Places-205 [98]: to compare the effect of the data distribution on pretraining, as in [53, 95]. We construct (b) by first removing all images with faces [91] and subsequently running our automated human and human part detection pipeline. The selected image IDs are provided in the supplementary material. We do not include manual verification because this dataset is only meant to measure the effect of removing humans from ImageNet. We also compare against supervised pretraining using datasets (a) and (b) for which labels are available.

**Frozen encoder evaluation.** Frozen encoder evaluations are seen as a gold-standard for evaluating representation learning performance due to their simplicity and ease of use. We include: the two standard evaluation datasets ImageNet [22] and Places [97], as well as ObjectNet [5], CIFAR-10 [55] as an example of a low-resolution dataset (though we note that its classes are contained within IN-1k), and the fine-grained classification datasets Oxford Flowers [66] and Herbarium 2019 [80]. We use linear probing, low-shot classification and clustering as tasks to evaluate the frozen encoders. For linear probing, we use MoCo-v2's default linear evaluation code, without any adjustments[6]. For low-shot classification we follow the public implementation of [58], again without tuning hyperparameters. For clustering, $K$-means is run on the embedded features of the validation set and their clustering is compared against the ground-truth labels, as in [73, 96]. Here $K$ is simply set to be equal to the number of ground-truth classes. This evaluation does not contain any hyperparameters.

**Finetuning evaluation.** For finetuning, we follow standard evaluation practices of the self-supervised literature: we evaluate object segmentation and detection. Beyond this, we also include human-focused efficient keypoint detection and pose estimation on COCO [60] to measure the effect of removing images with humans on person-centric tasks. Finally, we also finetune for object detection on PASCAL VOC [32] and long-tailed object segmentation on LVIS-v1 [40].

Specific details such as number of epoch, learning rates and schedules are provided in the Appendix. Additionally we include all code to reproduce the evaluations.

# 5  Results

When comparing pretraining datasets, a key factor is whether the pretraining and target datasets used for evaluation coincide or not. The realistic usage scenario is that they differ, which amounts to transfer learning. Yet, for benchmarking SSL methods, the non-transfer setting is often used instead. For example, both pretraining *and* evaluating on IN-1k does not test whether the features generalize beyond ImageNet. In the tables, we clearly distinguish these two cases by *graying the non-transfer learning results*. Naturally, there is a performance gap when training on **PASS** compared to matching pretraining and target datasets. However, for transfer learning we show that **PASS** performs on par to pretraining on ImageNet or similar datasets, even on downstream tasks involving humans, and for different SSL methods. Furthermore, we ablate the effect of using different pretraining splits and pretraining augmentations on downstream performances.

**Frozen encoder evaluation.** For the clustering evaluation in Table 2a, pretraining on **PASS** yields generalisable features that can outperform IN-1k and IN-1k$^*$ pretraining in the transfer learning

---

[6]For ObjectNet, which is a test set only, we transfer the linear classifier trained on IN-1k and report performance on the overlapping classes.

Table 2: **Frozen encoder evaluations.** We evaluate the encoders by (a) unsupervised clustering, (b) training a linear layer, and (c) training SVMs in a low-shot scenario on top of the frozen representations. Gray numbers indicate "non-transfer" evaluations on datasets that share all classes with the pretraining dataset and thus are expected to perform better than datasets with a different label distribution.

| Pretraining | IN-1k NMI | aRI | ObjN NMI | aRI | Places NMI | aRI | Flowers NMI | aRI |
|---|---|---|---|---|---|---|---|---|
| Random | 43.4 | 0.1 | 17.6 | 0.1 | 20.7 | 0.2 | 24.9 | 1.7 |
| Supv. IN-1k | 79.7 | 40.7 | 36.7 | 6.2 | 51.5 | 14.3 | 70.3 | 38.8 |
| Supv. IN-1k* | 75.9 | 34.8 | 33.1 | 4.3 | 47.0 | 10.7 | 70.4 | 39.0 |
| MoCo-v2 | | | | | | | | |
| on IN-1k | 64.0 | 12.7 | 24.3 | 1.1 | 44.2 | 7.5 | 62.1 | 30.1 |
| on IN-1k* | 63.2 | 12.2 | 23.7 | 1.0 | 42.4 | 6.4 | 62.5 | 32.0 |
| on Places | 54.7 | 5.5 | 22.8 | 0.8 | 51.1 | 12.0 | 55.2 | 21.6 |
| on PASS | 55.6 | 5.9 | 22.9 | 0.8 | 44.4 | 7.7 | 63.0 | 30.1 |

(a) **Clustering evaluation**: K-means is run on the validation sets and the assignments are compared against the ground-truth labels via the NMI and aRI (%).

| Pretraining | IN-1k | →ObjN | Plcs | C10 | Flwrs | H19 |
|---|---|---|---|---|---|---|
| Random | 4.2 | 0.4 | 16.6 | 14.4 | 9.0 | 3.1 |
| Supv. IN-1k | 76.3 | 27.3 | 51.5 | 74.0 | 90.7 | 44.0 |
| Supv. IN-1k* | 69.1 | 23.2 | 40.0 | 72.3 | 89.9 | 43.8 |
| MoCo-v2 | | | | | | |
| on IN-1k | 67.5 | 16.0 | 50.1 | 91.6 | 90.2 | 42.3 |
| on IN-1k* | 65.6 | 14.6 | 50.6 | 90.7 | 90.3 | 42.5 |
| on Places | 57.3 | 9.9 | 56.6 | 55.7 | 85.9 | 36.8 |
| on PASS | 59.5 | 11.7 | 52.8 | 81.0 | 89.2 | 46.5 |

(b) **Linear probing**: representations are used to train a linear layer using a standard schedule on a target dataset and top-1 accuracy is reported.

| | | Places | | | | | Pascal VOC07 | | | | | Herbarium19 | | | | |
|---|---|---|---|---|---|---|---|---|---|---|---|---|---|---|---|---|
| Pretraining | k= | 1 | 2 | 4 | 8 | 16 | 1 | 2 | 4 | 8 | 16 | 1 | 2 | 4 | 8 | 16 |
| Random | | 0.8 | 1.1 | 1.3 | 1.6 | 2.1 | 8.3 | 8.6 | 9.4 | 9.9 | 10.0 | 0.4 | 0.4 | 0.6 | 0.6 | 0.9 |
| Supervised IN-1k | | 15.3 | 21.3 | 27.0 | 32.3 | 36.4 | 54.0 | 67.9 | 73.8 | 79.7 | 82.3 | 4.8 | 7.4 | 11.5 | 16.7 | 22.6 |
| Supervised IN-1k* | | 12.7 | 17.5 | 22.9 | 27.9 | 32.2 | 48.6 | 62.1 | 68.7 | 75.2 | 78.8 | 4.8 | 7.8 | 11.8 | 17.2 | 23.2 |
| MoCo-v2 | | | | | | | | | | | | | | | | |
| on IN-1k | | 12.2 | 16.8 | 22.3 | 28.1 | 32.8 | 46.3 | 58.3 | 64.9 | 72.5 | 76.1 | 4.7 | 7.7 | 12.4 | 18.2 | 24.4 |
| on IN-1k* | | 11.0 | 15.4 | 20.4 | 25.9 | 30.6 | 43.2 | 55.3 | 61.9 | 69.5 | 73.8 | 4.8 | 7.5 | 12.0 | 17.7 | 24.4 |
| on Places | | 21.5 | 27.8 | 34.0 | 39.6 | 43.4 | 40.6 | 51.5 | 57.5 | 64.6 | 69.0 | 3.9 | 6.4 | 9.8 | 14.9 | 20.6 |
| on PASS | | 12.0 | 16.6 | 22.1 | 27.7 | 32.8 | 40.8 | 51.6 | 57.6 | 65.8 | 70.3 | 5.3 | 8.4 | 12.9 | 18.7 | 25.0 |

(c) **Low-shot classification** for k=1,2,4,8,16 images per class with training one-vs-all SVMs using 5 runs each. We report top-1 accuracy except for the multi-label VOC07, where we report mAP.

Table 3: **Finetuning representation evaluations.** We finetuning the encoders on (a) COCO2017 object detection & segmentation (see Appendix for Pascal VOC) and (b) dense human pose estimation. AP is COCO's AP on IoUs [0.5:0.95:0.05].

| | **Bounding-box** | | | | **Segmentation** | | | |
|---|---|---|---|---|---|---|---|---|
| Init. | AP | $AP_{50}$ | $AP_{75}$ | ⚥-AP | AP | $AP_{50}$ | $AP_{75}$ | ⚥-AP |
| Random | 26.4 | 44.0 | 27.8 | | 29.3 | 46.9 | 30.8 | |
| Supv. IN-1k | 38.9 | 59.6 | 42.7 | 54.9 | 35.4 | 56.5 | 38.1 | 47.1 |
| Supv. IN-1k* | 38.5 | 59.3 | 41.9 | 54.6 | 35.0 | 56.0 | 37.1 | 46.7 |
| MoCo-v2 | | | | | | | | |
| on IN-1k | 38.7 | 59.2 | 42.3 | 55.5 | 35.2 | 56.2 | 37.9 | 47.6 |
| on IN-1k* | 38.4 | 58.7 | 41.8 | 55.4 | 35.0 | 55.8 | 37.4 | 47.2 |
| on Places | 38.3 | 58.4 | 41.7 | 55.9 | 34.8 | 55.4 | 37.4 | 47.8 |
| on PASS | 38.0 | 58.5 | 41.5 | 55.5 | 34.7 | 55.4 | 37.1 | 47.5 |

| | **Dense-pose** | | **B-box** | **Seg.** |
|---|---|---|---|---|
| Init. | $AP_{GPS}^{dp}$ | $AP_{GPSm}^{dp}$ | $AP^{bb}$ | $AP^{sg}$ |
| Random | ——*does not train*—— | | | |
| Supv. IN-1k | 64.4 | 65.7 | 61.1 | 67.1 |
| Supv. IN-1k* | 64.2 | 65.5 | 60.9 | 66.9 |
| MoCo-v2 | | | | |
| on IN-1k | 65.0 | 66.3 | 61.7 | 67.6 |
| on IN-1k* | 64.8 | 66.1 | 61.4 | 67.0 |
| on Places | 64.9 | 66.0 | 61.8 | 67.1 |
| on PASS | 64.9 | 65.7 | 61.5 | 66.8 |

(a) Object detection & segmentation on COCO using R50-FPN, 1x schedule. ⚥-AP denotes AP on the person class.

(b) Dense human pose estimation with R50-FPN and the 's1x' schedule.

setting. The linear probing benchmark in Table 2b shows a similar pattern: **PASS** yields the best transfer performance on IN-1k, ObjectNet, Places205 and Herbarium19 (HT19). However, for CIFAR-10 transfer from ImageNet works better than Places and **PASS**. This is most likely due to the semantic overlap of these two datasets: all CIFAR-10 classes are contained in ImageNet, reducing the transfer gap.

Finally, we report the low-shot classification results in Table 2c. Here, we find that **PASS** pretraining performs close to IN-1k and surpasses IN-1k* when transferring to Places205. For PASCAL VOC, **PASS** is worse than both ImageNet variants, but outperforms Places205 pretraining. The gap to

Table 4: **Effect of SSL method.** We pretrain with recent SSL methods and evaluate: Linear probing and 4-shot classification (top-1 accuracy), PascalVOC (mAP), object detection COCO (mAP, ⚥ AP) and for PascalVOC (AP50). For clustering we report aRI.

| | Linear probing | | | | | Object detection | | 4-shot classific. | | | Clustering | |
|---|---|---|---|---|---|---|---|---|---|---|---|---|
| **SSL method** | IN-1k | ObjN | Plcs | Flwrs | HT19 | COCO (m,⚥) | PVOC | PVOC07 | Plcs | H19 | Plcs | Flwrs |
| MoCo-v2 | 59.5 | 11.7 | 52.8 | 89.2 | 46.5 | 38.0, 55.5 | 81.1 | 57.6 | 22.1 | 12.9 | 7.7 | 30.1 |
| MoCo-v2-CLD | 60.2 | 12.7 | 53.1 | 89.8 | 44.7 | 38.1, 54.7 | 81.2 | 59.6 | 23.0 | 13.9 | 7.9 | 31.4 |
| SwAV | 60.8 | 12.0 | 55.5 | 91.9 | 54.4 | 37.2, 52.3 | 72.9 | 61.9 | 23.0 | 16.0 | 9.4 | 36.7 |
| DINO (ViT-S16) | 61.3 | 11.9 | 54.6 | 92.2 | 48.5 | NA | NA | 64.3 | 24.9 | 18.2 | 10.8 | 43.1 |

Table 5: **Effect of pretraining augmentation.** Here we compare the effect of the minimum size parameter of the random-resized crop (RRC) augmentation when pretraining MoCo-v2.

| | Linear probing | | | | | Object detection | | 4-shot classific. | | | Clustering | |
|---|---|---|---|---|---|---|---|---|---|---|---|---|
| **RRC min. size** | IN-1k | ObjN | Plcs | Flwrs | HT19 | COCO (m,⚥) | PVOC | PVOC07 | Plcs | H19 | Plcs | Flwrs |
| 0.2 (default) | **59.5** | **11.7** | 52.8 | 89.2 | 46.5 | **38.0**, **55.5** | **81.1** | **57.6** | 22.1 | 12.9 | **7.7** | **30.1** |
| 0.1 (more zoom) | **59.5** | **11.7** | 53.4 | 89.6 | 48.9 | 37.8, 55.3 | **81.1** | **57.6** | 22.5 | 13.5 | **7.7** | 29.3 |

IN-1k pretraining might be due to the insufficient tuning of the SVM cost hyperparameter, as well as the semantic similarity of PascalVOC with ImageNet (common objects, object-centric images and indeed overlapping object categories). However, as shown below, different pretraining methods such as DINO can yield significant improvements. For Herbarium19, **PASS** yields the strongest pretraining performance, and is the only example where the self-supervised representation surpasses the supervised baselines for this benchmark.

**Finetuning evaluation.** Next we analyse the performance of the pretrained representations when they are used as initialisation for downstream tasks (c.f. Table 3). For Mask-RCNN based detection and segmentation on COCO (Table 3a, we find that the gaps between the different pretraining datasets are overall small and within $\pm1\%$ for the AP, AP50 and AP75 measures for both bounding-boxes (bb) and mask (mk) evaluations. We also report the performance on the 'person' object category (⚥) and find that the encoder pretrained on **PASS** still matches and surpasses the ones obtained with supervision on IN-1k and IN-1k*. In Table 3b we find similar trend for dense pose estimation, an inherently human-focused task, where the differences between pretraining datasets are even smaller. We provide further experimental results in the Appendix.

**Effect of data content.** In this section we study the effect of selecting different types of visual content, using the splits specified in Table 6. The first question is *whether humans matter or not* for pretraining. In the Humans split, we proceed as in **PASS** but skip the filtering step (in this case, 57% of the images contain humans and there is a 27% overlap with the filtered version). This leads to a small improvement in object detection 4-shot classification and a small decrease in performance for HT19, Flowers, and ObjectNet, but mostly staying within $\pm2\%$.

The second question is whether the distribution of *other* classes matters or not. We test biasing the selection of images by using an IN-1k pretrained classifier to match the class statistics of IN-1k as well as possible. The $\approx$IN800 split is obtained from the 6M images by retaining the most frequent 800 classes and retaining at most 2K samples per class, followed by subsequent sampling to a size of 1.28M. The $\approx$IN800 split leads to a large increase in performance for IN-1k and ObjectNet linear probing, echoing previous findings that the closeness of the pretraining distribution to the downstream task matters [53, 72]. However, it also leads to small decreases on datasets such as Places, H19 and Flowers.

The third question is whether capping the number of pictures per photographer matters. The Random6M split contains all 6M images left after cleaning and removing humans, as determined by pretrained models, but skipping the photographer balancing and manual cleanup step (the latter is for safety and only changes the dataset composition marginally). The Random split further restricts that to 1.28M. While the Random split generally performs less well than the other splits, Random6M achieves strong performances on many tasks. While this result is also partially due to the larger number of optimization steps (all methods were trained with 200 epochs), it might show how SSL methods can scale with more data.

Table 6: **Effect of pretraining data.** We report the same metrics as in 4.

| Data | Linear probing | | | | | Object detection | | 4-shot classific. | | | Clustering | |
|---|---|---|---|---|---|---|---|---|---|---|---|---|
| | IN-1k | ObjN | Plcs | Flwrs | HT19 | COCO (m,$\hat{x}$) | PVOC | PVOC07 | Plcs | H19 | Plcs | Flwrs |
| **PASS** | 59.5 | 11.7 | 52.8 | 89.2 | 46.5 | 38.0, 55.5 | 81.1 | 57.6 | 22.1 | 12.9 | 7.7 | 30.1 |
| • **Humans** | 59.7 | 11.2 | 53.5 | 87.7 | 45.3 | 38.2, 55.6 | 81.4 | 59.8 | 22.2 | 12.5 | 7.9 | 29.9 |
| ≈ **IN800** | 62.4 | 13.7 | 51.5 | 90.3 | 47.4 | 38.3, 55.2 | 81.7 | 61.6 | 21.4 | 12.8 | 7.5 | 31.2 |
| • **Random** | 58.7 | 11.2 | 52.7 | 88.6 | 46.6 | 37.8, 55.0 | 81.1 | 57.8 | 21.9 | 12.3 | 7.6 | 30.3 |
| • **Random6M** | 61.3 | 12.9 | 54.7 | 90.1 | 44.9 | 37.9, 55.3 | 81.0 | 61.3 | 24.4 | 11.9 | 8.4 | 31.3 |

Overall, **PASS** yields overall good performance without either containing humans, or the need for a pretrained classifier and is of size comparable to IN-1k.

**Effect of pretraining method.** Finally, we pretrain other self-supervised learning methods on our dataset and show results in Table 4. First, we compare MoCo-v2 against MoCo-v2-CLD [85], and find improvements similar in scale to the ones reported in [85]. Next, from the results of SwAV [12] and BYOL [38], we find that methods that performed better on IN-1k also tend to perform better on our dataset, show that **PASS** can be used instead of IN-1k to compare models.

**A note on augmentations.** In this paper we have solely used the hyperparameters that come with the methods, and as such, have been optimised by the computer vision community for the last few years on IN-1k. Augmentations play an important role for self-supervised learning [2, 14, 16], and strongly influence the final performance. We cannot replicate several years of augmentation tuning for our **PASS** dataset but can expect that with time, better settings can be found. To support this claim, we show one example of such adaption in Table 5 that we were able to find. By changing the minimum-size parameter used in random-resized crops from 0.2 to 0.1, we observe an improvement in performance in almost every benchmark. While small, the improvement might be explained by the fact that our dataset is less object-centric than ImageNet, and so stronger crops can be used, as there is no need for the crop to cover an object.

## 6 Discussion

In the introduction, we have motivated our new dataset **PASS** from a technical, ethical and legal perspective. By using CC-BY images, we greatly reduce the risk of using images in a manner incompatible with copyright. By avoiding the usage of search engines and labels to form a dataset, we avoid introducing corresponding biases. By excluding all images that contain humans, as well as other identifiable information and NSFW images, we significantly reduce data protection and other ethics risks for the data subjects. We have shown that, despite these changes, we can effectively pretrain neural networks using this data. By conducting extensive downstream task evaluations, we have shown that pretrained networks obtained using self-supervised training on this dataset are competitive to ImageNet on transfer settings and even on downstream tasks that involve humans.

However, several limitations remain. First, while we put care in filtering the images, automatically and manually, some harmful content might have slipped through.

Second, sampling images randomly from an uncurated large collection removes specific biases such as search engine selection but not others, for example the geographic bias. Furthermore, we added one significant bias: there are no people in these pictures, despite the fact that a large fraction of all images in existence contain people. While this appears to be acceptable for model *pretraining*, **PASS** cannot be used to learn models of people, such as for pose recognition.

Thirdly, since **PASS** contains no labels, **PASS** (in contrast to ImageNet) cannot be used alone for training and benchmarking. For this, curated datasets remain necessary, which continue to carry many of the issues of privacy and copyright described in the paper.

Despite these limitations, we believe that **PASS** is an important step towards curating and improving our datasets to reduce ethical and legal risks for many tasks and applications and at the same time to challenge the SSL community with a new, more realistic training scenario of utilizing images not obtained from a labeled dataset.

## Acknowledgements

We are thankful to Abhishek Dutta and Ashish Thandavan for their great support. We thank Rajan and his team of annotators at Elancer for their precise work. We are grateful for support from the AWS Machine Learning Research Awards (MLRA), EPSRC Centre for Doctoral Training in Autonomous Intelligent Machines & Systems [EP/L015897/1], the Qualcomm Innovation Fellowship, a Royal Society Research Professorship, and the EPSRC Programme Grant VisualAI EP/T028572/1. C. R. is supported by Innovate UK (project 71653) on behalf of UK Research and Innovation (UKRI) and by the European Research Council (ERC) IDIU-638009.

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

# Appendix

## Table of Contents

## A   Dataset Access

During review, we had provided instructions for the reviewers to view the dataset in the OpenReview submission. The dataset is hosted by Zenodo [31] and can be found under this URL:

https://zenodo.org/record/5501843

## B   Image attributions

The creators for the images in Fig. 1, are (top-left to bottom-right): *RichTatum, chongeileen, Vlad Iorsh, millstastic, Ross_Angus, nfeli777, semper_fi_brother, jj-photo, BluEyedA73, Pazit Polak, florianplag, Johan Lange, kimadababe, Alzheimer's Association - Greater Illinois, Esme_Vos, babeltravel, Henrique B Costa, Dude of Lego, matthewreid, Russ Dill, theunwiseman, Wenkan, at8eqeq3, Larry1732, Haroldo Kennedy, DGriebeling, Tez_kuma, Overman Alawami, FreeCat, fczuardi, Norisa1, pescatello, jasonlsraia, semanticwebcompany, jumblejet, Rain San Martin, crackerbelly, Problemkind, M_Hartman Photography, antaean, andyket, mirven, Quinn Rossi,*

*MARCO_QUARANTOTTI, -Tripp-, JulieHagenbuch, frank drewett, MeRyan, Sharib4rd, Schlusselbein2007, BBR1245, Officer Phil, cadyellow, Andreas Joensen, FuFuWolf, zerojay, Spoon Monkey, Tatters80, gypsygirl.photography, ChristinaT, Irene Vlachou, allenreichert, rs-foto, Sara Cimino, JonoMueller, Lock The Gate, m01229, ebis50, Speculum Mundi, fontosiskola, rakh1, dreamcicle19772006 ON OFF .*

All images in this paper and the dataset are licensed by the **CC-BY** (`https://creativecommons.org/licenses/by/2.0/`) license and contain information on their creator. On the above url, this license is described as follows:

2. Remix, transform, and build upon the material for any purpose, even commercially.

3. This license is acceptable for Free Cultural Works.

4. The licensor cannot revoke these freedoms as long as you follow the license terms.

## C  Dataset Generation Details

### C.1  Automated pipeline

For the detection of faces we use RetinaFace [23] and the MobileNet0.25 model from the PyTorch RetinaFace repository (`https://github.com/biubug6/Pytorch_Retinaface`). We keep all parameters, including the detection threshold value of $0.02$. For the automated detection of humans and human parts we use the highly performant Cascade-RCNN trained using the 3x schedule on COCO, available from the detectron2 library [87]. The threshold for detections is set to its default value of $0.5$.

Finally, before passing the images to the professional human annotators, we run the a model pretrained to detect "not safe for work" (NSFW) content on the set of images from ([57]). After visually inspecting the top 500 entries for both the 'hentai' and the 'porn' categories, and not finding any positive results, we pass the data to the human annotators.

### C.2  Human verification

In Fig. 5, we provide two screenshots of the verification instructions given to the annotators. We have manually assessed the quality of these annotations using two independent samples of size 1000 each and only found 15 false positives and 2 minor false negatives. We informed the annotation team of our findings to further improve the quality. The important error type here are false negatives, as this would mean humans or personal information remains undetected.

After human verification 2% of images are flagged and subsequently removed. The vast majority of flagged images stem from the presence of car licence plates.

### C.3  Removal of duplicates

We also conduct a basic removal of duplicates process using the Find Identical Images tool [26]. Using this we find and remove 2529 self-duplicates, 114 duplicates with MS COCO, and 9 duplicates with ImageNet LSVRC-12. We further checked against duplicates with Places-205, but found none.

## D  Implementation Details

### D.1  Representation learning experiments

**MoCo-v2.** We pretrain the MoCo-v2 models using its default settings for 200 epoch using the publicly available repository (`https://github.com/facebookresearch/moco`). This includes training using the cosine learning rate schedule, augmentations consisting of random resized crops, colorjitter, random horizontal flipping and blurring with a batch-size of 256 and a learning rate of $0.03$. On two RTX6000 GPUs this training requires around one week to complete.

Figure 5: **Human verification instructions.** We have blurred faces and other personally identifiable information solely for in these screenshots. The professional annotators saw unblurred versions to provide a clearer understanding of what images require flagging.

**SwAV.** We pretrain SwAV [12] using the fixed hyperparameters for 200 epoch pretraining with a batch-size of 256 found at the publicly available repository (`https://github.com/facebookresearch/swav`). For this we use 4 RTX6000 GPUs and this requires training for a bit more than a week. SwAV's augmentations include those of MoCo-v2 and in addition Multi-Crop, in which in addition to the two 224-sized crops, six 96x96-sized crops are extracted and used for computing the loss.

**DINO.** We pretrain DINO [13] using for 100 epochs with a batch-size of 256 using the publicly available repository (`https://github.com/facebookresearch/dino`). Training augmentations include those of SwAV, including Multi-Crop, as well as solorization, introduced in [38]. Here, the architecture trained is ViT-B [24] and training takes around 2 days on 8 GPUs.

**Pretrained models.** We utilise pretrained models for the MoCo-v2 models trained on Places and ImageNet from the MoCo-v2 repository and from the repository of [95]. In addition we use PyTorch's publicly available models: a supervisedly pretrained ResNet50 model for benchmarking and a pretrained MobileNet-v3 for generating the ≈IN800 split.

### D.2 Downstream tasks

**Cluster evaluation.** We conduct these experiments on the test sets of the datasets by resizing the images to smaller-side 256pixels and taking a centered 224x224 crop. These images' embeddings after global average pooling are computed and input in k-means, implemented by the FAISS [49] library. For k-means, we use 50 iterations and 5 restarts and set K to be equal to the number of ground-truth classes. NMI, aRI calculations are computed by scikit-learn [69].

**Linear probing.** We exactly follow MoCo-v2's implementation. For ease, we describe it here: Training is done for 100 epochs on the global average pooled features of the encoders without the use of an additional BatchNorm layer. The learning rate is dropped by a factor of 0.1 at epochs 60, 80 starting at a value of 30.0. The batchsize is 256, and we use a single GPU. The augmentations during training consist of random horizontal flipping and RandomResizedCrops to size 224x224 with the

Table 7: Differences between linear evaluation implementations.

| model | Linear eval code | |
| --- | --- | --- |
| | SwAV | MoCo-v2 |
| SwAV | 60.4 | 57.5 |
| MoCo-v2 | 47.0 | 59.1 |

(a) Pretraining on **PASS**.

| model | Linear eval code | |
| --- | --- | --- |
| | SwAV | MoCo-v2 |
| SwAV | 72.7 | 69.2 |
| MoCo-v2 | 58.2 | 67.7 |

(b) Pretraining on IN-1k.

Table 8: SwAV linear probing performance. We report the linear evaluation performance of the SwAV pretrained model when using SwAV's linear evaluation code.

| SSL method | Linear probing | | | | |
| --- | --- | --- | --- | --- | --- |
| | IN-1k | ObjN | Plcs | Flwrs | HT19 |
| SwAV | 60.4 | 10.4 | 55.0 | 88.3 | 38.7 |

default parameters (crops of sizes (0.08, 1.0) of the original size and variations in aspect ratio of (3/4, 4/3)). For linear probing on the Vision Transformer [24], we use the 100 epoch linear evaluation code of [13].

**Data-efficient SVM classification.** We exactly follow the publicly available implementation of PCL [58] (`https://github.com/salesforce/PCL`). Here, an one-vs-all SVM is fit on the data using a cost parameter $c = 0.5$, which we keep at its default value. The features for the SVM are obtained by smaller-side-resizing to 256 and taking a center crop of 224x224.

**Finetuning.** We follow the publicly available implementation of MoCo [44]. For the R50-C4 architecture used in the Pascal VOC experiments, this includes the use of an additional normalisation layer and training for 24k steps. For FPN architectures (used for COCO detection, segmentation, keypoint, densepose and LVIS segmentation) this includes the use of SyncBatchNorm and training using the '1x' schedule for which trains for 90K steps and has a learning rate warmup for the first 1000 steps from 0.02/100 and subsequent training with an initial learning rate of 0.02 being divided by 10 at steps 60K and 80K. For the data-efficient keypoint detection we use the R50-FPN with a fixed split containing 5% of the annotations of COCO2017's train split, train for 9K steps and evaluate on the full validation split. For the dense pose estimation on COCO [60] we use an R50-FPN with chart-based embeddings [39] and the 's1x' schedule, which decays the learning rate at steps 100K, 120K and trains for 130K steps in total. Training is done on the 2014 densepose training split and the 2014 'valminusminival' split, and performance tested on the 2014 'minival' split. Note that the default '1x' schedule of LVIS includes 180K iterations, so in these terms our LVIS training can be referred to as '0.5x'. Instead of a batch-size of 16 we use a batch-size of 8 on two GPUs and divide the learning rates by half and the multiply the learning-rate schedules by two. We use the detectron2 library [87].

Following the MoCo-v2 implementation and experiments, we evaluate object detection and segmentation on COCO [60] using a Mask-RCNN [43] with the R50-FPN backbone [61] with the '1x' schedule and object detection on Pascal VOC07+12 [32] using a R50-C4 using 24k steps.

### D.2.1 A note on evaluation hyperparameters

Finally, we note that comparing different self-supervised methods using linear evaluation is non-trivial, as the optimal settings (learning rate and schedule) can depend on the encoder's pretraining method. As we show in Table 7, both MoCo-v2 and SwAV suffer strongly when instead of their linear evaluation code, the other's is used. For completeness, we report the linear probing numbers of SwAV when their linear evaluation code is used in Table 8.

While outside of this work, frozen evaluation of methods which do not require further finetuning and are close to deployment also include the clustering evaluation in the paper (which can be combined with a Hungarian algorithm to yield label predictions), kNN evaluations (as done in e.g. [13]) and the normalisation of feature statistics before training the linear layer, as in [52].

### D.2.2 Dataset details

For detection and segmentation in Table 3a we use COCO2017, which contains 80 labeled objects with segmentation masks and boxes. The training set contains 118K images with 850K annotations, and the validation set 5K images with 36K annotations.

For dense human pose estimation in Table 3b, COCO2014 (a subset of COCO2017) is used, which contains masks of people and their bodies' 3D surface on a mesh. The training and validation sets contain 32K and 1.5K images with 99K and 25K persons instances, respectively.

For object detection in Table 9a we use the common Pascal VOC "07+12" split containing 20 object categories. This split uses the VOC7's test set (2.5K images and 15K annotations) for evaluation and merges the two trainval sets for finetuning (16.5K images and 47K annotations).

For data-efficient keypoint detection in Table 9b, we generate a random, fixed split of COCO2017, resulting in 1K images with 2K annotations for training and the full test set of 5K images with 10K annotations.

For detection and segmentation in Table 9c, we use LVIS-v1 [40], which contains 1203 labeled objects with segmentation masks and boxes. The training set contains 100K images with 1.2M annotations, and the validation set 5K images with 244K annotations.

For cross-domain transfer in Table 10, we use multiple smaller-scale datasets taken from the Visual Task Adaptation Benchmark (VTAB) [93]: CLEVR-count [50] contains synthetic rendered images of objects and the task is to count the number of objects via 8-way classification and contains 70K, 15K images for training and testing. Our construction of the dataset follows the code from the VISSL repository [37]. DTD [19] contains 47 classes of different describable textures (*e.g.* 'honeycombed' or 'sprinkled'), and we use the first train and test-split supplied in the paper, containing around 2K images each. Both Eurosat [45] and Resisc45 [18] are remote sensing image classification datasets with 10 and 45 classes respectively (*e.g.* 'baseball diamond', or 'river'). They do not come with a natural train/test split, so we create a fixed training split using 60% and 20% of the data, following VTAB [93]. This yields around 21K and 25K images for training and 5K and 6K for testing, for Eurosat and Resisc45, respectively.

## E Additional Experimental Results

Table 9: **Finetuning representation evaluations.** a) PascalVOC07+12 detection and (b) human keypoint estimation and c) Object segmentation on LVIS. AP denotes COCO's AP on IoUs [0.5:0.95:0.05].

| Init. | Bounding-box | | |
| --- | --- | --- | --- |
| | $AP_{50}$ | AP | $AP_{75}$ |
| Random | 52.5 | 28.1 | 26.2 |
| Supv. IN-1k | 81.3 | 53.5 | 58.8 |
| Supv. IN-1k* | 80.9 | 53.5 | 58.8 |
| MoCo-v2 | | | |
| on IN-1k | 82.3 | 56.8 | 63.1 |
| on IN-1k* | 81.5 | 55.7 | 62.0 |
| on Places | 81.5 | 56.3 | 62.8 |
| on **PASS** | 81.1 | 55.5 | 61.2 |

(a) Object detection on Pascal VOC07+12 using Faster-RCNN, R50-C4, and 24K steps.

| Init. | $AP^{kp}$ | $AP^{kp}_{50}$ | $AP^{kp}_{75}$ |
| --- | --- | --- | --- |
| Random | 25.9 | 52.9 | 21.5 |
| Supv. IN-1k | 39.7 | 69.4 | 39.5 |
| MoCo-v2 on **PASS** | 40.0 | 68.7 | 39.8 |

(b) Human keypoint detection on 5% of COCO.

| Init. | AP | $AP_r$ | $AP_f$ | $AP_c$ |
| --- | --- | --- | --- | --- |
| Random | 11.4 | 2.7 | 18.3 | 8.7 |
| Supv. IN-1k | 18.9 | 7.6 | 26.3 | 16.6 |
| MoCo-v2 on **PASS** | 16.6 | 4.1 | 24.9 | 14.2 |

(c) Object segmentation on LVIS-v1.

### E.1 Object detection on PascalVOC

In Table 9a we report the performances of object detection using R50-C4 on Pascal VOC with 24k steps (as in MoCo). As reported in previous works [44, 95], the self-supervised baselines outperform supervised pretraining especially the more strict AP and AP75 measures. We find that pretraining on **PASS** falls within 1% of the performance of pretraining on IN-1k without humans.

## E.2 Data-efficient keypoint detection on COCO

In Table 9b, we show the results of finetuning the encoders using a small sample of ground-truth annotations from the COCO 2014 keypoint annoations. While using 100% of the data results in almost no gains between pretrained and random initialisations (as shown in [44]), we see a large difference when only 5% of the annotations are considered. We find that pretraining on **PASS** yields the strongest keypoint detection results in terms of AP and AP75, surpassing even supervised IN-1k pretraining.

## E.3 Long-tailed instance segmentation on LVIS-v1

In Table 9c, we show the results of finetuning for object segmentation using Mask-RCNN with R50-FPN on LVIS-v1 [40], which contains 1203 classes. We find that different to object detection on COCO in Table 3a, the supervisedly trained baseline performs best with an overall AP of $18.9$, and the MoCo-v2 pretrained on **PASS** yielding an AP of $16.6$, and large gap in the 'rare' categories, as measured by $AP_r$.

## E.4 Cross-domain transfer

Table 10: **Cross-domain transfer**. Linear-probing Top-1 accuracies are reported for various datasets whose domain is further from typical web-images.

| Model | Resisc45 | Clevr-cnt | Eurosat | DTD |
|---|---|---|---|---|
| sup. IN-1k | 90.2 | 61.8 | **96.6** | 64.5 |
| MoCo-v2 Places | 91.0 | 70.4 | 93.5 | 62.9 |
| MoCo-v2 IN-1k | 90.7 | 72.6 | 96.5 | 66.7 |
| MoCo-v2 **PASS** | **91.4** | **73.1** | 95.4 | **66.8** |

In Table 10, we report cross-domain transfer results on the following datasets and tasks taken from VTAB [93]: CLEVR-count [50] (object counting), DTD [19] (texture classification) and Eurosat [45] (satellite image land-cover classification) Resisc45 [18] (remote sensing image classification). We use the same linear-evaluation protocol as in the remainder of the paper, i.e. the settings from MoCo-v2, without any hyperparameter tuning.

From Table 10 we find that MoCo-v2 trained on our **PASS** dataset does very well across those datasets, surpassing both IN-1k supervised and MoCo-v2 on IN-1k for three out of four cases. This shows that pretraining on **PASS** is a viable strategy even if the downstream task domain is very different.

# F Dataset Documentation: Datasheets for Datasets

Here we answer the questions outlined in the datasheets for datasets paper by Gebru et al. [33].

## F.1 Motivation

**For what purpose was the dataset created?**   Neural networks pretrained on large image collections have been shown to transfer well to other visual tasks where there is little labelled data, i.e. transferring a model works better than starting with a randomly initialized network every time for a new task, as many visual features can be repurposed. This dataset has as its goal to provide a safer large-scale dataset for such pretraining of visual features. In particular, this dataset does not contain any humans or human parts and does not contain any labels. The first point is important, as the current standard for pretraining, ImageNet and its face-blurred version only provide pseudo-anonymity and furthermore do not provide correct licences to the creators. The second point is relevant as pretraining is moving towards the self-supervised paradigm, where labels are not required. Yet most methods are developed on the highly curated ImageNet dataset, yielding potentially non-generalizeable research.

**Who created the dataset (e.g., which team, research group) and on behalf of which entity (e.g., company, institution, organization)?**   The dataset has been constructued by the research group "Visual Geometry Group" at the University of Oxford at the Engineering Science Department.

**Who funded the creation of the dataset?**   The dataset is created for research purposes at the VGG research group. Individual researchers have been funded by AWS Machine Learning Research Awards (MLRA), EPSRC Centre for Doctoral Training in Autonomous Intelligent Machines & Systems [EP/L015897/1], the Qualcomm Innovation Fellowship, Innovate UK (project 71653) on behalf of UK Research and Innovation (UKRI) and by the European Research Council (ERC) IDIU-638009.

## F.2 Composition

**What do the instances that comprise the dataset represent (e.g., documents, photos, people, countries)?**   This dataset only contains photos. In addition we provide tabular meta-data for these images, which contain information such as the creator's username and image capture date.

**How many instances are there in total (of each type, if appropriate)?**   The dataset contains 1.4M images, resulting in 181GB as a tar file.

**Does the dataset contain all possible instances or is it a sample (not necessarily random) of instances from a larger set?**   The dataset is a sample of a larger set—all possible digital photographs. As outlined in Section 3 we start from an existing dataset, YFCC-100M, and stratify the images (removing images with people and personal information, removing images with harmful content, removing images with unsuitable licenses, each user contributes at most 80 images to the dataset). This leaves 1.6M images, out of which we take a random sample of 1.28M images to replicate the size of the ImageNet dataset. While this dataset can thus be extended, this is the set that we have verified to not contain humans, human parts and disturbing content.

**What data does each instance consist of?**   Digital photographs uploaded by users of the flickr platform.

**Is there a label or target associated with each instance?**   No. Our dataset deliberately does not contain labels.

**Is any information missing from individual instances?**   Not from the dataset. Note however that the meta-data that we additionally provide is not complete and might have non-uniform missing values.

**Are relationships between individual instances made explicit (e.g., users' movie ratings, social network links)?**   Not applicable: each image stands on its own and we do not provide relationships between these.

**Are there recommended data splits (e.g., training, development/validation, testing)?**   As outlined in the intended usecases, this dataset is meant for pretraining representations. As such, the models derived from training on this dataset need to be evaluated on *different datasets*, so called down-stream tasks. Thus the recommended split is to use all samples for training.

**Are there any errors, sources of noise, or redundancies in the dataset?** No.

**Is the dataset self-contained, or does it link to or otherwise rely on external resources (e.g., websites, tweets, other datasets)?** No. The dataset contains links to the publicly hosted mirror of the YFCC dataset on Amazon Web Services.

**Does the dataset contain data that might be considered confidential (e.g., data that is protected by legal privilege or by doctor-patient confidentiality, data that includes the content of individuals' non-public communications)?** No.

**Does the dataset contain data that, if viewed directly, might be offensive, insulting, threatening, or might otherwise cause anxiety?** No. Besides checking for human presence in the images, the annotators were also given the choice of flagging images for disturbing content, which once flagged was removed.

**Does the dataset relate to people? If not, you may skip the remaining questions in this section.** No.

**Does the dataset identify any subpopulations (e.g., by age, gender)?** NA

**Is it possible to identify individuals (i.e., one or more natural persons), either directly or indirectly (i.e., in combination with other data) from the dataset?** NA

**Does the dataset contain data that might be considered sensitive in any way (e.g., data that reveals racial or ethnic origins, sexual orientations, religious beliefs, political opinions or union memberships, or locations; financial or health data; biometric or genetic data; forms of government identification, such as social security numbers; criminal history)?** NA

### F.3 Collection process

**How was the data associated with each instance acquired?** The data was collected from the publicly available dataset YFCC-100M which is hosted on the AWS public datasets platform. We have used the meta-data, namely the copyright information to filter only images with the CC-BY licence and have downloaded these using the aws command line interface, allowing for quick and stable downloading. In addition, all files were subsequently scanned for viruses using Sophos SAVScan virus detection utility, v.5.74.0.

**What mechanisms or procedures were used to collect the data (e.g., hardware apparatus or sensor, manual human curation, software program, software API)?** Our dataset is a subset of the YFCC-100M dataset. The YFCC-100M dataset itself was created by effectively randomly selecting publicly available images from flickr, resulting in approximately 98M images.

**If the dataset is a sample from a larger set, what was the sampling strategy (e.g., deterministic, probabilistic with specific sampling probabilities)?** See the similar question in the Composition section.

**Who was involved in the data collection process (e.g., students, crowdworkers, contractors) and how were they compensated (e.g., how much were crowdworkers paid)?** As described, the data was collected automatically by simply downloading images from a publicly hosted S3 bucket. The human verification was done using a professional data annotation company that pays 150% of the local minimum wage.

**Over what timeframe was the data collected?** The images underlying the dataset were downloaded between March and June 2021 from the AWS public datasets' S3 bucket, following the download code provided in the repo. However the images contained were originally and taken anywhere from 2000 to 2015, with the majority being shot between 2010-2014.

**Were any ethical review processes conducted (e.g., by an institutional review board)?** No.

**Does the dataset relate to people? If not, you may skip the remainder of the questions in this section.** No, this is one of the specific purposes of this dataset.

### F.4 Preprocessing/cleaning/labeling

**Was any preprocessing/cleaning/labeling of the data done (e.g., discretization or bucketing, tokenization, part-of-speech tagging, SIFT feature extraction, removal of instances, processing of missing values)?** After the download of approx. 17M images, the corrupted, or single-color images were removed from the dataset prior to the generation of the dataset(s) used in the paper. The images were not further preprocessed or edited.

**Was the "raw" data saved in addition to the preprocessed/cleaned/labeled data (e.g., to support unanticipated future uses)?** Yes. The creators of the dataset maintain a copy of the 17M original images with the CC-BY licence of YFCC100M that sits at the start of our dataset creation pipeline.

**Is the software used to preprocess/clean/label the instances available?** We have only used basic Python primitives for this. For the annotations we have used VIA [27, 28]

### F.5 Uses

**Has the dataset been used for any tasks already?** In the paper we show and benchmark the intended use of this dataset as a pretraining dataset. For this the dataset is used an unlabelled image collection on which visual features are learned and then transferred to downstream tasks. We show that with this dataset it is possible to learn competitive visual features, without any humans in the pretraining dataset and with complete license information.

**Is there a repository that links to any or all papers or systems that use the dataset?** We will be listing these at the repository.

**What (other) tasks could the dataset be used for?** We believe this dataset might allow researchers and practitioners to further evaluate the differences that pretraining datasets can have on the learned features. Furthermore, since the meta-data is available for the images, it is possible to investigate the effect of image resolution on self-supervised learning methods, a domain largely underresearched thus far, as the current de-facto standard, ImageNet, only comes in one size.

**Is there anything about the composition of the dataset or the way it was collected and preprocessed/cleaned/labeled that might impact future uses?** Given that this dataset is a subset of a dataset that randomly samples images from flickr, the image distribution is biased towards European and American creators. As in the main papers discussion, this can lead to non-generalizeable features, or even biased features as the images taken in other countries might be more likely to further reflect and propagate stereotypes [84], though in our case these do not refer to sterotypes about humans.

**Are there tasks for which the dataset should not be used?** This dataset is meant for research purposes only. The dataset should also not be used for, e.g. connecting images and usernames, as this might risk de-anonymising the dataset in the long term. The usernames are solely provided for attribution.

### F.6 Distribution

**Will the dataset be distributed to third parties outside of the entity (e.g., company, institution, organization) on behalf of which the dataset was created?** No.

**How will the dataset will be distributed (e.g., tarball on website, API, GitHub)?** The dataset will be provided as a csv along with code hosted on GitHub that allows the user to download the images in our dataset. In addition, we hope to also host it as a single tarball on our servers.

**When will the dataset be distributed?** Starting from July 2021.

**Will the dataset be distributed under a copyright or other intellectual property (IP) license, and/or under applicable terms of use (ToU)?** CC-BY.

**Have any third parties imposed IP-based or other restrictions on the data associated with the instances?** No.

**Do any export controls or other regulatory restrictions apply to the dataset or to individual instances?** Not that we are are of. Regular UK laws apply.

### F.7 Maintenance

**Who is supporting/hosting/maintaining the dataset?** The dataset is supported by the authors and by the VGG research group. The main contact person is Yuki M. Asano. We host the dataset on zenodo: https://zenodo.org/record/5528345.

**How can the owner/curator/manager of the dataset be contacted (e.g., email address)?** The authors of this dataset can be reached at their e-mail addresses: *{yuki,chrisr,vedaldi,az}@robots.ox.ac.uk*. In addition, we have added a contact form in which we can be contacted anonymously at https://forms.gle/tkZugt2DJnFdCE1i6.

**Is there an erratum?** If errors are found and erratum will be added to the website.

**Will the dataset be updated (e.g., to correct labeling errors, add new instances, delete instances)?** Yes, updates will be communicated via the website. The dataset will be versioned.

**If the dataset relates to people, are there applicable limits on the retention of the data associated with the instances (e.g., were individuals in question told that their data would be retained for a fixed period of time and then deleted)?** Not applicable.

**Will older versions of the dataset continue to be supported/hosted/maintained?** If even after our verification we find further images that contain humans or problematic content, we will remove those further images from existing splits to preserve the goal of this dataset.

**If others want to extend/augment/build on/contribute to the dataset, is there a mechanism for them to do so?** Others are free to reach out to us if their ideas can build on this dataset. All code will be made available.

### F.8 Other questions

**Is your dataset free of biases?** No. There are many kinds of biases that can either be quantified, e.g. geo-location (most images originate from the US and Europe) or camera-model (most images are taken with professional DSLR cameras not easily affordable), there are likely many more biases that this dataset does contain. The only thing that this dataset does not contain are humans and parts of humans, as far as our validation procedure is accurate.

**Can you guarantee compliance to GDPR?** No, we cannot comment on legal issues.

### F.9 Author statement of responsibility

The authors confirm all responsibility in case of violation of rights and confirm the licence associated with the dataset and its images.

