# OpenReview forum: "PASS: An ImageNet replacement for self-supervised pretraining without humans"
_NeurIPS.cc/2021/Track/Datasets_and_Benchmarks/Round1 — NeurIPS 2021 Datasets and Benchmarks Track (Round 1)_

### Official Review · Reviewer_gQLQ · 2021-07-04
**This work is of great value and may contribute to the unsupervised learning community.**

**Rating:** 10
**Confidence:** 4
**Correctness:** Yes, all top-standard.
**Clarity:** Clear and easy to follow

**Strengths:**

(1) From a technical, ethical, and legal perspective, the proposed PASS dataset is very significant and hopefully becomes an ImageNet replacement for self-supervised pretraining.
(2) This paper has a complete description of the data generation process, including descriptions of the original dataset, the experimental setup, software used, objects involved, instructions to creators, methods for quality control, and so on.
(3) Different self-supervised approaches train well on the PASS dataset, yielding strong image representations.
(4) The experimental results show that pre-trained networks obtained using self-supervised training on the PASS dataset are competitive to the ImageNet dataset on most downstream tasks.
(5) The authors deeply analyze the limitations of this work.



**Weaknesses:**

(1) In Table 2c, PASS is significantly worse than both ImageNet variants on the Pascal VOC07 dataset. An explanation of this phenomenon should be added.
(2) More experimental details in Table 3 should be provided. For example, the number of used annotations, the training details, and so on.
(3) The PASS dataset does not contain humans or body parts at all. Why can PASS pretraining achieve comparable performance to ImageNet pretraining on downstream tasks that involve humans such as human pose estimation?

**Additional Feedback:**

n/a

**Documentation:**

yes

**Ethics:**

no, all discussed in the paper

**Relation To Prior Work:**

Yes, very comprehensive

**Summary And Contributions:**

This paper proposed an unlabelled PASS dataset for self-supervised pretraining. PASS only contains images with CC-BY license and complete attribution metadata to address the copyright issue. Besides, all images on the PASS dataset do not contain humans or body parts at all to address the data protection issue. The authors conduct an extensive downstream task evaluation to discuss performance differences when SSL methods are trained using ImageNet and PASS. This work is of great value and may contribute to the unsupervised learning community.

---

> ### Author Response · Authors · 2021-07-09
> **Response**
>
> We thank the reviewer for their in-depth review and their truly encouraging feedback.
> Here we address the raised points.
>
> 1) >In Table 2c, PASS is significantly worse [...] An explanation of this phenomenon should be added.
>
> We will have added the following text to the revised version of the paper:
>
> *For PASCAL VOC, PASS is worse than both ImageNet variants, but outperforms Places205 pretraining. The gap to IN-1k pretraining might be due to the insufficient tuning of the SVM cost hyperparameter, as well as the semantic similarity of PascalVOC with ImageNet (common objects, object-centric images and indeed overlapping object categories). However, as shown below, different pretraining methods such as DINO can yield significant improvements even on this task.*
>
>
>
> 2) > More experimental details in Table 3 should be provided. For example, the number of used annotations, the training details, and so on.
>
> Thank you for pointing this out. While the finetuning training details are provided in Appendix D and referred to in the sentence just before Sec. 5 ("*Specific details such as number of epoch, learning rates and schedules are provided in the Appendix. Additionally we include all code to reproduce the evaluations.*"), we have now added further details regarding the dataset and annotation sizes to our revised version in appendix D2.2.
>
>
>
>
> 3) >The PASS dataset does not contain humans or body parts at all. Why can PASS pretraining achieve comparable performance to ImageNet pretraining on downstream tasks that involve humans such as human pose estimation?
>
> We believe this is mainly because of two reasons: First, since the network is fully finetuned, it might be able to adapt existing features with the available training data, even if it does not have exact representations of humans or human parts. Second, it might also be the case that detection of human pose does not require "high level” semantics, and that indeed detection of general visual features such as edges and shapes is enough for these tasks.

---

### Official Review · Reviewer_ATJm · 2021-07-04
**A new image dataset that is diverse and does not have privacy issues.**

**Rating:** 6
**Confidence:** 3
**Correctness:** Yes, these claims are correct to the …
**Clarity:** Yes.

**Strengths:**

1. The authors have constructed PASS very carefully to avoid ethical problems. All images in the PASS dataset has CC-BY license and no copyright issue. In addition, the dataset contains no images of people and has filtered potentially problematic images.

2. The author has conducted extensive evaluations, e.g., training with MoCo-v2, SwAV and DINO. These experiments show that models trained on PASS can generate competitive image representations as those trained on ImageNet / Places205.

**Weaknesses:**

1. The scale of PASS (which has not been labeled) is only comparable to ImageNet-1K (which has been labeled). Given that It is easier to collect a large number of unlabeled images, I would expect that the size of PASS should be much larger than ImageNet-1K. Otherwise, researchers won't evaluate on this dataset given the wide adoption of ImageNet-1K and no clear performance advantage in training with PASS.


**Additional Feedback:**

Given that PASS is unlabelled and ImageNet-1K is labelled, it is misleading to call it "an ImageNet replacement" and I would suggest the author to change the title.

--- Post Rebuttal ---

I've read the author's rebuttal and will keep my score unchanged. I still think that it is confusing to call PASS an "ImageNet replacement" because these two datasets have different focuses (self-supervised v.s. supervised) and cannot be directly compared.


**Documentation:**

Yes.

**Ethics:**

Avoiding any ethical issues is the main focus of PASS. The author spent lots of efforts in avoiding potential ethical and copyright problems. Thus, there is no ethical concerns.

**Relation To Prior Work:**

Yes.

**Summary And Contributions:**

The paper proposed the PASS dataset, which has the same number of images as ImageNet-1K and only contains non-human images with CC-BY license. Self-supervised learning experiments show that PASS is diverse and can learn good image representations.

---

> ### Author Response · Authors · 2021-07-09
> **Response**
>
> We thank the reviewer for taking the time to review the paper and their comments. We respond to the points raised in the following.
>
> > Given that it is easier to collect a large number of unlabeled images, I would expect that the size of PASS should be much larger than ImageNet-1K.
>
> It is true that it is easier to collect and verify a large unlabeled dataset. Indeed, it would be possible to conduct human verification on the larger 6M images of Tab 6 (last row). However, as we find in that table, this increases performance only in some metrics while requiring almost 5x as much training. This slow gain from increasing the size of pretraining data has also been found in [1,2] and indeed the first MoCo paper [3] and motivates the usage of medium-sized datasets for developing pretraining methods. In addition, by limiting the size to that of ImageNet-1K, adoption of existing methods, and training schedules is made as easy, transparent and comparable as possible.
>
>
>
>
> > Given that PASS is unlabelled and ImageNet-1K is labelled, it is misleading to call it "an ImageNet replacement" and I would suggest the author to change the title.
>
> Thank you for the suggestion. Since the full title of the paper is “An ImageNet replacement for self-supervised pretraining“, the “replacement” aspect merely refers to the self-supervised pretraining use case. In addition, we hope that our explanation of the motivation for our unlabeled dataset in the title, abstract, introduction and discussion is precise enough to avoid confusion. If there is consensus among the reviewers that the title is confusing, we are happy to change it.
>
>
> **References**
>
> [1] Goyal et al. Self-supervised Pretraining of Visual Features in the Wild. 2021.
>
> [2] Cole et al. When Does Contrastive Visual Representation Learning Work?. 2021.
>
> [3] He et al. Momentum Contrast for Unsupervised Visual Representation Learning. CVPR 2020.

---

### Official Review · Reviewer_j6RL · 2021-07-04
**A self-supervised pretraining dataset with more ethical and privacy considerations**

**Rating:** 6
**Confidence:** 3

**Strengths:**

1. The paper proposes a reasonable alternative to ImageNet with less privacy, license, and ethical issues.
2. The paper establishes a sensible pipeline for curating a relatively clean dataset from internet images, which could benefit future efforts in building datasets.
3. The paper points out by experiment that current methods have their hyperparameters tailored to ImageNet pretraining, and shows that to achieve better performance on different pretraining datasets, efforts are needed to tune the parameters.


**Weaknesses:**

1. Currently pretraining on PASS still underperforms ImageNet pretraining on the important segmentation and detection task by approximately 1%, which is a gap enough to make people reconsider using PASS. However I do not attribute the gap solely to the lack of persons in PASS, and I believe a major issue is that the hyperparameters are tuned for ImageNet. This is shown in table 6 row 3 where simply matching the class distribution improves detection result by 0.5%. Re-tuning the parameters could probably remedy this problem, but it would require significant efforts. I believe currently this dataset will be most useful mainly when license and privacy are the main concerns. Mainstream adoption might be difficult unless the performance on these important downstream tasks could match ImageNet pretraining.

2. Personally, I believe the current manual filtering is a bit too aggressive, especially in removing all “license plates, signatures, or handwriting” from the images. I understand the rationale of protecting the information, however, it may not be necessary to remove basically all recognizable texts from the dataset as it may influence downstream task performance that requires this knowledge. I would recommend the authors to test their modal pre-trained on PASS in a cross-domain transfer setting, e.g. VTAB[1], to ensure the cross-domain generalizability is not harmed by the strict curation.

[1] A Large-scale Study of Representation Learning with the Visual Task Adaptation Benchmark, 2019


**Additional Feedback:**

Please refer to the weakness section for comments and suggestions.


**Clarity:**

The writing of the paper is clear and the theme of improving ethics and privacy is explained well.


**Correctness:**

The claims are valid in this paper. The dataset construction and evaluation are sound.


**Documentation:**

The dataset is promised to be released and the evaluation is clear and sounds reproducible.


**Ethics:**

This paper improves upon previous work in terms of solving ethics issues.


**Relation To Prior Work:**

The related work is well introduced in this paper.


**Summary And Contributions:**

This paper introduces PASS, a dataset for pretraining self-supervised representation learning methods. Compared to ImageNet-1k, this dataset has several benefits:

1. All images follow the CC_BY license hence no license issue.
2. There is no human in the datasets, eliminating a lot of privacy concerns.
3. The author makes an effort to remove undesirable or biased images from the dataset.

The authors apply several state-of-the-art self-supervised representation learning methods on this dataset and evaluated the performance on multiple downstream tasks. They conclude that the performance is comparable to pretraining with ImageNet-1k.

---

> ### Author Response · Authors · 2021-07-09
> **Response**
>
> We thank the reviewer for their time and feedback on our paper. In the following, we will respond in detail to the raised points:
>
> 1) >PASS still underperforms ImageNet pretraining on the important segmentation and detection task [...]. Mainstream adoption might be difficult unless the performance on these important downstream tasks could match ImageNet pretraining.
>
> As the reviewer also notes (“I believe a major issue is that the hyperparameters are tuned for ImageNet”), we think that this slight underperforming is likely to be solved in subsequent works and via larger-scale tuning of hyperparameters. *In addition*, as we describe in the paper, we believe that adoption of new datasets is -- and should not be --- solely because of superior performance, but that instead multiple factors play a role: for industry and academia using images without obtaining a suitable license can be risky. Additionally, ethical problems of using personal data cannot be ignored. We thus strongly believe that this dataset is a major step in the right direction for the community.
>
>
>
> 2) > Personally, I believe the current manual filtering is a bit too aggressive, especially in removing all “license plates, signatures, or handwriting” from the images. [...] it may not be necessary to remove basically all recognizable texts from the dataset as it may influence downstream task performance [...]. I would recommend the authors to test their modal pre-trained on PASS in a cross-domain transfer setting, e.g. VTAB.
>
> While we do remove license plates and other personal identifiable information from the images because it constitutes personal identifiable information (e.g. according to GDPR), contrary to the reviewers comment, we do not remove _all_ texts. For example in the dataset browser provided to the reviewers, we have several examples of writing just on the first page, such as a cardboard advertisement, a street sign or writing on t-shirts.
>
> Regarding “cross-domain transfer”: most of our evaluations already contain a domain shift, but in the short time for the discussion period, we will conduct additional experiments that are further from the typical internet images and taken from VTAB: CLEVR-count (counting), DTD (texture classification) and Eurosat (satellite image classification). We will post another reply once these experiments have finished.

---

> > ### Author Response · Authors · 2021-07-12
> > **Update: results on cross-domain transfer**
> >
> > As promised in our first response, we have now run further linear evaluations (using the exact same protocol as in the other parts of the paper) on the following datasets and tasks: CLEVR-count (counting), DTD (texture classification) and Eurosat & Resisc45 (satellite image classification).
> > The results are shown in the table below, which we will add (along with the dataset details) to the Appendix in our paper.
> >
> > | **Model**      | Resisc45 | Clevr-count | Eurosat | DTD  |
> > |----------------|----------|-----------|---------|------|
> > | Supervised IN-1k     | 90.2     | 61.8      | **96.6**    | 64.5 |
> > | MoCo-v2 on Places | 91.0     | 70.4      | 93.5    | 62.9 |
> > | MoCo-v2 on IN-1k  | 90.7     | 72.6      | 96.5    | 66.7 |
> > | MoCo-v2 on PASS   | **91.4**     | **73.1**      | 95.4    | **66.8** |
> >
> > As we can see from the table, MoCo-v2 trained on our PASS dataset does very well across those datasets,  surpassing both IN-1k supervised and MoCo-v2 on IN-1k for three out of four cases. This shows that pretraining on PASS is a viable strategy even if the downstream task domain is very different.

---

> > > ### Comment · Reviewer_j6RL · 2021-07-12
> > > **Thanks for the nice cross-domain results**
> > >
> > > Thanks to the authors for the updated cross-domain results. I am glad to see in these cross-domain settings PASS outperforms IN-1k. Although the current dataset has slightly weaker performance on the detection and segmentation task, I do believe it could bring a meaningful contribution to the community. So I will change my evaluation accordingly.

---

### Author Response · Authors · 2021-07-14
**Post discussion phase update**

We thank the reviewers again for their time and feedback. On day 2 after the reviews came out we have provided clarifications and initial responses to all reviewers. In the following, we provide a brief overview of the changes to the paper during the discussion phase:

a) Based on the feedback of reviewer *gQLQ*, we have added further details on the downstream datasets utilised, as well as further explanations of some results. This yielded the "revision 1" of the paper.

b) Based on the feedback of reviewer *j6RL*, we have additionally run linear probing on cross-domain transfer tasks of DTD, CLEVR-count, Eurosat and Resisc45, where we surpass ImageNet pretraining in 3/4 cases. The results and experimental details were added to the paper in "revision 2".

We believe the paper is now even stronger and look forward to providing the community with this new pretraining dataset.

---

### Decision · Program_Chairs · 2021-07-26

**Decision:**

Accept

**Comment:**

All reviewers recommended accept. AC didn't find any reason to overturn the consensus recommendation.